# OpenReview forum: "Language Models Struggle to Explain Themselves"
_ICLR.cc/2024/Conference — Submitted to ICLR 2024_

### Official Review · Reviewer_ZS9G · 2023-10-22

**Soundness:** 2 fair
**Presentation:** 2 fair
**Contribution:** 2 fair
**Rating:** 5
**Confidence:** 3

**Summary:**

This paper attempts to investigate whether LLMs can generate faithful explanations for their internal processes. To do so, authors claim that using behavioral/blackbox tests with adversarial inputs suffices to imply that the model is using a `rule`. The authors later build on this assumption to test whether the model’s explanation aligns with the suggestion that the model is using that rule. Under these assumptions, authors quantify the degree to which the models' explanations are faithful to their decision-making processes.

**Strengths:**

1. The problem that LLMs struggle to provide faithful explanations for their decision is an important one, pronounced also in the existing literature (e.g. Turpin et al.). This paper differs in that the authors attempt to provide a methodology and a dataset to evaluate the faithfulness of the explanations, which could be a meaningful contribution if executed well.

2. If the authors' assumption that the model is indeed using the rule holds true (which I am skeptical about, see W1 and Q3); the remainder of the evaluations become interesting. In that, if we can verify that a model is using a rule, we can use consistency tests to understand whether LLMs’ explanations are faithful to their decision-making processes; and this would be a useful methodology.

**Weaknesses:**

1. The connection between an explanation and hypothesizing that the model uses a rule is extremely brittle, in my opinion. I will state my understanding here, and please correct me if I am wrong.
- The authors suggest that if a model attains high accuracy in a task, and is robust to adversarial perturbations, then the model is using a rule.
- The above is too strong of an assumption, and `Therefore the goal is to test if the model is as close as feasible to using the known rule` jump the authors make is too strong. A model can exploit another rule, that is also invariant to the perturbation distribution chosen by the authors, yet is different than the ground truth rule.
- To their justice, the authors themselves also recognize this, `could be using the more complex rule “contains the word ‘lizard’ or ‘zyrpluk”` and I certainly agree – the behavioral results need not imply that a model is using a rule. This strong assumption makes the remainder of the results more brittle, in my view.
- Overall, I believe the verification of this strong assumption should extend beyond having a handful of perturbations to the existing prompts and needs more justification. This constitutes the most significant limitation in the paper, since if the assumption does not hold the paper’s methodology becomes invalidated.

2. The freeform articulation test appears brittle to me. Even though an articulation does not match the ground truth rule designed in the test, if the articulated rule could also explain the same set of examples, then it would not be correct to label this as an inaccuracy, in my opinion. There may be multiple correct strategies, and the model may be using it. This comment is related to W2.

3. The above two weaknesses are also made more brittle by the low sample size ($<256$). This also makes the finetuning results somewhat brittle – authors finetune models with billions of parameters with tens of examples, and this alone may contribute to why finetuning does not improve articulation accuracy.

**Questions:**

1. In Section 2.2, description of the task, authors say `5 space-separated lower case words`, yet Figure 2 contains 3 words, although the appendix includes 5-word examples. If 5 words are used across the board, I would either fix Figure 2 or add a disclaimer to avoid confusion.

2. Section 2 immediately jumps to defining tasks, but until reading the rest of the paper I could not understand what these tasks are for (for explanations? For measuring model accuracy? For both?). I would add a short paragraph to the start of Section 2 to clarify what we are defining in the coming tasks.

3. I would be curious to see whether the articulated explanations that are labeled as incorrect could also explain the provided examples. Specifically, what fraction of the ~30% of freeform evaluations provided by GPT-4 that are not aligned with the ground truth rule could also explain the provided in-context examples? This could be an easy test that could be evaluated by another LLM in the loop (give the articulation, and ask whether it could explain the in-context example). Alternatively, this could also be evaluated by hand since it will be a few tens of examples.

Minor

1. (Intro, Paragraph 1) I’m not sure what AGI Alignment means in this specific context, personally I believe if we use such desiderata for systems in scientific papers, we at least need to define what it is.

2. (Intro, Last Paragraph) authors cite: `as well as GPT-4 (Bubeck et al., 2023)`; but Bubeck et al. hardly is the paper that introduced GPT-4. I’d cite the technical report by OpenAI, since the model was released by them, not by Bubeck et al.

---

> ### Author Response · Authors · 2023-11-19
> **Responses to your helpful feedback**
>
> First of all, thank you for your feedback and for explicitly laying out your understanding of the paper’s methodology. We’ve addressed the minor improvements. We would like to clarify a few key points that may have been misunderstood, as outlined below.
>
> > “The authors suggest that if a model attains high accuracy in a task, and is robust to adversarial perturbations, then the model is using a rule.”
>
> We think that this is a misunderstanding. We’d like to clarify that we’re not claiming that the model’s using the rule if it describes the model’s behavior on a range of in- and out-of-distribution inputs, rather that the model’s behavior is closely approximated by the rule. Therefore, if the model were to articulate the rule, it would be an accurate description of its behavior, since it describes the model’s behavior on a wide range of in- and out-of-distribution inputs.
>
> Section 2.4 has been updated to clarify this point.
>
> > To their justice, the authors themselves also recognize this, could be using the more complex rule “contains the word ‘lizard’ or ‘zyrpluk” and I certainly agree – the behavioral results need not imply that a model is using a rule. This strong assumption makes the remainder of the results more brittle, in my view.
>
> Just for clarity and to reiterate the point above: we’re not assuming that the model is using the rule “contains the word ‘lizard’” when it could well be using “contains the word ‘lizard’ or ‘zyrpluk”. If both rules accurately describe the model’s behavior on a wide range of in- and out-of-distribution inputs, articulating either would be marked as correct on the freeform articulation task.
>
> Indeed, we designed the multiple choice articulation task with edge cases like this in mind to avoid them altogether; only one option is consistent with the data, as mentioned in Section 2.4. In the freeform articulation task, we controlled for this by manually reviewing 200 freeform articulations for the in-context evaluation (out of 480, and mainly GPT-3/4's since they were the only models with reasonable articulations), and similarly hundreds for the finetuning evaluation. We empirically found that models either articulated the rule R we expected, or articulated another rule R’ that failed to be consistent with the few-shot examples entirely. That is, out of hundreds of manually reviewed freeform articulations, the model never articulated a rule R’’ which was consistent with the few-shot examples but semantically different to R (we did however see rephrasings of R, which our language model discriminator was good at catching, as demonstrated in Appendix B.4.1).
>
> We’ve added the details of our manual review process in Appendix B.3, including a spreadsheet of notes we took while manually reviewing articulations.
>
> > “Overall, I believe the verification of this strong assumption should extend beyond having a handful of perturbations to the existing prompts and needs more justification. This constitutes the most significant limitation in the paper, since if the assumption does not hold the paper’s methodology becomes invalidated.”
>
> It was stated that we used “a handful of perturbations” to test whether a model classifies according to a rule. However, as we state in the paper, we used 15 diverse adversarial attacks. These extend beyond simple perturbations and include “instruction injection” and using LLMs to generate adversarial inputs. Thus, if a model achieves ~100% accuracy in matching the intended simple rule under held-out adversarial attacks, we think it’s reasonable to say that its behavior on a diverse range of out-of-distribution examples is very well approximated by the simple rule. Still, this is a quantitative difference (15 attacks vs. “a handful”) and we want to address the qualitative point the reviewer makes.

---

> ### Author Response · Authors · 2023-11-19
> **Responses to your helpful feedback (continued)**
>
> > “The freeform articulation test appears brittle to me. Even though an articulation does not match the ground truth rule designed in the test, if the articulated rule could also explain the same set of examples, then it would not be correct to label this as an inaccuracy, in my opinion. There may be multiple correct strategies, and the model may be using it. This comment is related to W2.”
>
> > I would be curious to see whether the articulated explanations that are labeled as incorrect could also explain the provided examples. Specifically, what fraction of the ~30% of freeform evaluations provided by GPT-4 that are not aligned with the ground truth rule could also explain the provided in-context examples? This could be an easy test that could be evaluated by another LLM in the loop (give the articulation, and ask whether it could explain the in-context example). Alternatively, this could also be evaluated by hand since it will be a few tens of examples.
>
> Just to reiterate the point made above: indeed, we agree this is possible and have accounted for it in both the multiple choice and freeform articulation tasks. In practice, we didn’t find a single instance of this happening in hundreds of manually reviewed freeform articulations, and avoided this edge case entirely by construction in the multiple choice articulation task.
>
> > The above two weaknesses are also made more brittle by the low sample size (< 256). This also makes the fine tuning results somewhat brittle – authors finetune models with billions of parameters with tens of examples, and this alone may contribute to why finetuning does not improve articulation accuracy.
>
> We think this is an important misunderstanding. We trained GPT-3 on 100’s of classification and articulation demonstrations, not 10’s. We also ensured our approach was consistent with OpenAI’s best practices for fine-tuning [1]. However, the training set _is_ generated by 10’s of rule functions which is a limitation (see the “Increase the variety of articulations” dot point in Section 3 and “Finetuning on freeform articulations” in Section 4).
>
> [1] https://platform.openai.com/docs/guides/legacy-fine-tuning/classification
>
> > In Section 2.2, description of the task, authors say 5 space-separated lower case words, yet Figure 2 contains 3 words, although the appendix includes 5-word examples. If 5 words are used across the board, I would either fix Figure 2 or add a disclaimer to avoid confusion.
>
> Thanks for raising this potential confusion. Yes, we always use 5 words and have modified Figure 2 to contain 5 words.
>
> > Section 2 immediately jumps to defining tasks, but until reading the rest of the paper I could not understand what these tasks are for (for explanations? For measuring model accuracy? For both?). I would add a short paragraph to the start of Section 2 to clarify what we are defining in the coming tasks.
>
> Thanks for this suggestion. We agree this would be useful, and have added such a paragraph.

---

> > ### Comment · Reviewer_ZS9G · 2023-11-19
> > **Response to the Rebuttal**
> >
> > Thank you for your detailed rebuttal.
> >
> > > **Clarifications around my understanding**
> >
> > I appreciate your explanations, yet I am still unclear about the claims in the paper. In several places of the manuscript, e.g.
> > > *Overall, GPT-3 performs poorly at articulating the rules it follows* (Section 1 last paragraph)
> >
> > > *We show that it entirely fails to articulate the simple rules it uses* (Section 3.2.3, first paragraph)
> >
> > > *GPT-3-c-f’s ability to articulate its classification rule in natural language* (Section 3.2.3, third paragraph)
> >
> > > *In this paper, we create a dataset, ArticulateRules, to evaluate how well LLMs can articulate the rule they’re using* (Section 6, first paragraph),
> >
> > the authors suggest the evaluation is to articulate the rules that a model follows. However, in the rebuttal text, they suggest
> >
> > > *we’re not claiming that the model’s using the rule if it describes the model’s behavior on a range of in- and out-of-distribution inputs, rather that the model’s behavior is closely approximated by the rule*.
> >
> > Isn't there a mismatch here? In particular, authors suggest they evaluate whether a rule approximates a behavior, yet they discuss whether or not the model is using a rule, 'its classification rule', `performs poorly at articulating the rules it follows`. According to the claim you suggest, shouldn't the conclusions be around `performs poorly at articulating the rules that closely approximate its behavior`? Otherwise, I do find the former to be an incorrect statement, based on what the authors suggest their claim is. This imprecision in language causes significant confusion and reduces the reliability of claims, in my opinion.
> >
> > If I'm missing something here, I would be happy to reconsider.
> >
> > > **Controlling for different rules that can still explain the behavior**
> >
> > Thank you for the clarification here and the addition to the paper with Appendix C.3, I appreciate it. It is interesting to see that (using the authors' notation) there is never a case where there exists $R'' \neq R$ that the model produces that can still approximate the behavior.
> >
> >
> > > **Low Sample Size**
> >
> > I disagree that there is a misunderstanding here. The number of parameters is orders of magnitude larger than the training data size, and the important weakness is that the low number of samples makes the experiments less conclusive (and it is not that one chooses to call it "tens" or "hundreds"). I'm not sure how the OpenAI documentation the authors shared addresses this question. In particular, what do the authors think a reviewer should find in the legacy finetuning documentation page? If there is an important point, please do share it and I will consider it.

---

> ### Author Response · Authors · 2023-11-21
> **Response**
>
> Thank you for your quick and insightful reply.
>
> > Isn't there a mismatch here? In particular, authors suggest they evaluate whether a rule approximates a behavior, yet they discuss whether or not the model is using a rule, 'its classification rule', performs poorly at articulating the rules it follows. According to the claim you suggest, shouldn't the conclusions be around performs poorly at articulating the rules that closely approximate its behavior? Otherwise, I do find the former to be an incorrect statement, based on what the authors suggest their claim is. This imprecision in language causes significant confusion and reduces the reliability of claims, in my opinion.
>
> Thank you for raising this point; we appreciate you clearly stating your concerns with examples. We acknowledge that the current phrasing could be confusing and have updated the relevant sections in the paper. For example, “Overall, GPT-3 performs poorly at articulating the rules it follows” was changed to “Overall, GPT-3 performs poorly at articulating rules which closely approximate its behavior”.
>
> > I disagree that there is a misunderstanding here. The number of parameters is orders of magnitude larger than the training data size, and the important weakness is that the low number of samples makes the experiments less conclusive (and it is not that one chooses to call it "tens" or "hundreds"). I'm not sure how the OpenAI documentation the authors shared addresses this question. In particular, what do the authors think a reviewer should find in the legacy finetuning documentation page? If there is an important point, please do share it and I will consider it.
>
> Thank you for your response and apologies for the lack of specificity in the prior reply. Specifically, we were referring to the “Classification” header in the OpenAI docs (https://platform.openai.com/docs/guides/legacy-fine-tuning/classification), which states to “Aim for at least ~100 examples per class”, and the "Conditional generation" header (https://platform.openai.com/docs/guides/legacy-fine-tuning/conditional-generation), which states to "Aim for at least ~500 examples". The non-legacy finetuning version of the docs (https://platform.openai.com/docs/guides/fine-tuning/example-count-recommendations) also states “We recommend starting with 50 well-crafted demonstrations and seeing if the model shows signs of improvement after fine-tuning. In some cases that may be sufficient…” Indeed, our experimental findings agreed with these recommendations; we tried finetuning on both 100’s and 1000’s of demonstrations, and found 100’s to be sufficient (i.e. it achieved a comparable validation accuracy) for both the classification and articulation tasks, as mentioned in Section 3.2.

---

> > ### Comment · Reviewer_ZS9G · 2023-11-21
> > **Response to authors**
> >
> > Thank you for your rebuttal! I will be revising my score based on the discussion.

---

> > > ### Author Response · Authors · 2023-11-22
> > >
> > > Thank you for your thorough, clear and fast feedback too; we really appreciate it!

---

### Official Review · Reviewer_8rGm · 2023-10-31

**Soundness:** 3 good
**Presentation:** 2 fair
**Contribution:** 2 fair
**Rating:** 5
**Confidence:** 4

**Summary:**

The paper investigates whether autoregressive LLMs can give faithful high-level explanations of their own internal processes. To explore this, the paper introduces a dataset, ArticulateRules, and a test bed. These include few-shot text-based classification tasks generated by simple rules and free-form explanation generation, as well as multiple choice selection of explanations. The evaluation focuses on GPT-3 LLMs with different parameter sizes and GPT-4.

**Strengths:**

- The authors construct a test bed along with a dataset to evaluate interesting and novel directions of explainable AI in the context of recent autoregressive LMs.
- The scope of the paper is clearly described. In general, is the paper well written, including clear descriptions of the dataset and introduced tasks.
- Datasets and testbeds provide the foundation for future investigations and therefore a valuable contribution.

**Weaknesses:**

- The paper has a large emphasis on tuning GPT-3 in order to increase the performance in „self-explain“ tasks. Therefore, the authors miss the opportunity to provide deeper insights into why the models fail on the introduced test bed and the self-explain tasks. More details on why the models fail would strengthen the paper.
	- While the selected definition of self-explain is interesting, the paper ignores feature attribution methods. However, attribution maps and their correlation with the LMs output would further support the insights.
	- Authors decided to introduce their test bed by evaluating black-box LMs. However, models with white box access (e.g., Pythia [https://arxiv.org/abs/2304.01373]) or Llama would provide more insights, e.g., by investigating input attribution. But even with only black-box access, input feature attribution can be generated, e.g., using perturbation methods or Contrastive Input Erasure [https://arxiv.org/pdf/2202.10419.pdf]. Further open LMs would provide greater reproducibility.
- Tables and figures exceed the paper margins

- The paper states, "We evaluate to what extent LLMs are capable of producing faithful explanations of their own behavior via finetuning." However, the faithfulness of the explanation is largely ignored in the present evaluation. See e.g. https://arxiv.org/pdf/1911.03429.pdf.

Minor Comments:
- Sec 2.2.1 seems to be the only subsection of 2.2. It also seems to be misplaced. It rather fits the 2.4. Same for 2.1. In Section 2, the authors could distinguish between the test-bed, including the tasks, and the dataset.
- "Curie showed the first signs of life." Try to avoid such phrases.
- Remove figure references in the description of the main finding in the introduction since they interrupt the reading flow. Place Figure 2 as the first figure since it summarizes the methodology of the paper quite well and is more helpful than Figure 1 at the beginning of the paper. Current Figure 1 should be placed in the results.
- Captions should be left aligned

**Questions:**

While the insights provided in the paper are interesting, it is not clear to me what the benefit of self-explaining capabilities as defined in the paper is. As you briefly discuss in the limitation section, if a model can self-explain itself as the present paper defines it, it is still unclear if these explanations are faithful, i.e., corresponding to the reasons used in the model's internal process. Same for the opposite case. If it can not self-explain as defined in the paper, it does not prove that it is actually not using the right reasons. It would be interesting to see a relation between, e.g., the quality of feature attribution-based explanations and the investigated self-explain capabilities of a model.

---

> ### Author Response · Authors · 2023-11-19
> **Responses to your helpful feedback**
>
> Thanks for this detailed review. We are glad you see the value in releasing benchmarks of this form, and find our contribution valuable, well scoped, and clearly explained. We’ve updated figures to be within the paper margins and addressed all minor comments. We address the remaining weaknesses below.
>
> > “The paper states, "We evaluate to what extent LLMs are capable of producing faithful explanations of their own behavior via fine tuning." However, the faithfulness of the explanation is largely ignored in the present evaluation. See e.g. https://arxiv.org/pdf/1911.03429.pdf”
>
> Thanks for suggesting this paper (https://arxiv.org/pdf/1911.03429.pdf) which we have added to related work. This paper studies faithfulness in rationales, a specific form of explanation. A rationale is considered faithful if it actually informs model predictions. One way to measure is to manipulate an input to remove the rationale (“comprehensiveness”).
>
>
> We understand “faithfulness” in a similar way. An articulation of a rule is faithful if it actually informs the model’s classifications on held-out examples. If a model articulates a rule R but is shown to classify held-out examples consistent with a distinct rule S, then this articulation is necessarily an unfaithful explanation. This is what we actually find for GPT-3. Specifically, GPT-3 produces freeform and multiple choice articulations of rules that do not match its classification behavior on held-out examples. Thus, GPT-3 fails to be faithful, even after finetuning.
>
> GPT-4, by contrast, achieves significantly higher accuracy in articulating rules that match its classification behavior (~70% top-3 freeform articulation accuracy for GPT-4 vs ~10% for GPT-3). This suggests that GPT-4 may be “capable of producing faithful explanations”, although it still fails on ~30% of rules. As we discuss in the limitations section, there is a question of whether even 100% accurate articulation of rules is sufficient condition for faithfulness. It could be that the articulated rule describes the model’s behavior but does not causally influence its predictions. There are various approaches to testing this (e.g. manipulating internal representations in whitebox models). We think these are important questions for future work in this area.
>
> We hope we have clarified how our results relate to faithfulness. We have amended the paper to include these points about faithfulness in the introduction, conclusion and results sections.
>
> > “More details on why the models fail would strengthen the paper.”
>
> Thanks for this suggestion. We are not able to definitively explain why models fail to perform well, but are able to speculate. One speculation is that models seem poorly incentivized by both pretraining and fine tuning to produce faithful explanations of their own behavior, in which case it’s unsurprising that models struggle to generalize in this way, which our work demonstrates. However, we agree that explaining why models fail at this task is an interesting direction, and we’ve added this to our future work section.
>
> > While the insights provided in the paper are interesting, it is not clear to me what the benefit of self-explaining capabilities as defined in the paper is. As you briefly discuss in the limitation section, if a model can self-explain itself as the present paper defines it, it is still unclear if these explanations are faithful, i.e., corresponding to the reasons used in the model's internal process. Same for the opposite case. If it can not self-explain as defined in the paper, it does not prove that it is actually not using the right reasons. It would be interesting to see a relation between, e.g., the quality of feature attribution-based explanations and the investigated self-explain capabilities of a model.
>
> Thanks for pointing this out. Self-explanation is a desirable property for models to have, in general. As we noted above, we view high accuracy on our benchmark to be a necessary, but not sufficient test of language model self explanation. We chose such an approach as procedurally generated black box behavioral evaluations provide a cheap, fast and scalable test of model capabilities, while still being informative.
>
> We agree that studying the relation between explanation capabilities and feature attribution-based methods would be informative. We have listed this as a possible area of future work.

---

> ### Author Response · Authors · 2023-11-19
> **Responses to your helpful feedback (continued)**
>
> > Authors decided to introduce their test bed by evaluating black-box LMs. However, models with white box access (e.g., Pythia [https://arxiv.org/abs/2304.01373]) or Llama would provide more insights, e.g., by investigating input attribution. But even with only black-box access, input feature attribution can be generated, e.g., using perturbation methods or Contrastive Input Erasure [https://arxiv.org/pdf/2202.10419.pdf]. Further open LMs would provide greater reproducibility.
>
> We agree the corresponding experiments on white box models would be interesting. We plan on releasing our dataset so that such experiments can be run by others in the community. In this work, we elected to first evaluate OpenAI models to determine where the current SOTA articulation capabilities were.
>
> However, in response to similar comments by other reviewers, we are in the process of running some evaluations on the Llama series of models, results for which should be ready in the next few days. We will share the results of these in a top level comment.
>
> We also take the point that white box methods could be illuminating. We too thought so, and discussed some potential avenues in this vein in the future work sections. We chose not to explore these due to time constraints, and these experiments falling out of the scope of our key contributions. Thanks for the comment on perturbation methods - these too would be interesting to explore, and we have added this to our future work section.
>
> > “In Section 2, the authors could distinguish between the test-bed, including the tasks, and the dataset.”
>
> Thanks for this. We have added a paragraph at the start of section 2 explaining on a high level the tasks and datasets.
>
> > Tables and figures exceed the paper margins
>
> Thanks for raising this point. We have fixed all of these issues.
>
> > Minor Comments
>
> Thanks. We have also fixed all of these stylistic issues.

---

> ### Author Response · Authors · 2023-11-21
>
> Dear Reviewer 8rGm, once again thank you for your time and effort in providing your thoughtful review of our paper. As we are now entering the last day or so of the rebuttal period we wanted to just reach out to see if there was any feedback from our response -- we appreciate that this will be a busy time for you but we hope that our current response has addressed any current concerns and are keen to engage further if there are any outstanding concerns. Best wishes, the Authors

---

> > ### Comment · Reviewer_8rGm · 2023-11-22
> >
> > Thank you for the clarifications and for updating the paper accordingly. Including them already improved the paper. While additional experiments on white box models intended to open the opportunity of generating input influence-based explanations and in turn draw further conclusions, I still appreciate the extra effort.
> > I don't have any remaining questions.

---

### Official Review · Reviewer_ZENG · 2023-11-05

**Soundness:** 3 good
**Presentation:** 2 fair
**Contribution:** 2 fair
**Rating:** 5
**Confidence:** 4

**Summary:**

The paper introduces ArticulateRules, a dataset of few-shot text classification tasks with associated simple rule explanations, to evaluate whether large language models can provide faithful high-level explanations of their internal processes behind competent classification. The authors test a range of models in-context and find articulation accuracy increases with model size, especially from GPT-3 to GPT-4. However, even finetuned GPT-3 fails to articulate explanations matching its classifications, though it shows some capability on easier multiple choice tasks. Overall, the analysis indicates that current large language models struggle to provide high-level self-explanations, though abilities are emerging in GPT-4. The dataset provides a useful benchmark for future work on testing and improving self-explanation in large language models.

**Strengths:**

1. This paper works on a timely and important topic, i.e., whether LLMs can give faithful high-level explanations of their own internal processes.
2. This paper introduces a novel dataset, ArticulateRules, that provides a concrete way to test whether large language models can explain their reasoning and decision making processes behind text classifications.
3. This work comprehensively evaluates a range of large language models in-context, showing a clear correlation between model size/capability and articulation accuracy.
4. The experimental analysis demonstrates specifically that even a very large model like GPT-3 fails completely at articulating explanations for a significant portion of simple rules. This highlights major limitations of current self-explanation abilities.

**Weaknesses:**

1. The major concern is that the authors claim or speculate that GPT-4 has over 175 billion parameters. However, even during the paper review period, the exact size of GPT-4 remains unclear.
2. Even if it is confirmed that GPT-4 has over 175 billion parameters, this alone does not lead to the conclusion that "articulation accuracy increases with model size." The reason is that the differences between the models shown in Figure 1 are not only in their number of parameters, but also in other significant factors like training data. Simply comparing model sizes does not account for these other variables that likely also impact articulation accuracy.
3. The ArticulateRules dataset is relatively small and simple, focusing on text classification tasks based on simple syntactic rules. Performance on this limited dataset may not reflect abilities on more complex real-world tasks.
4. Only one main model architecture (GPT-3/4) is tested in detail. Other model types may have different explanation capabilities that are not explored.

**Questions:**

Please refer to the weaknesses section

---

> ### Author Response · Authors · 2023-11-19
> **Responses to your helpful feedback**
>
> Many thanks for an insightful review. We’re glad that you find our results to highlight a major limitation of current models and the topic to be a timely one. We are additionally pleased you found our in-context experiments to be comprehensive and the scaling trend to be informative. We address the weaknesses below.
>
> > Simply comparing model sizes does not account for these other variables that likely also impact articulation accuracy.
>
> Agreed; this is a good point and is well-taken. We’ve replaced the claim about articulation accuracy increasing with model size with a more nuanced claim which acknowledges that multiple factors other than model size (e.g. the training set or model architecture) play a role in the the increased general capabilities of GPT-4, and of the observed increase in articulation accuracy. The motivation behind studying a series of models in this manner was to establish whether articulation capabilities would arise by default with scale.
>
> > The major concern is that the authors claim or speculate that GPT-4 has over 175 billion parameters. However, even during the paper review period, the exact size of GPT-4 remains unclear.
>
> Thanks for noting this. This is a good point and such speculation is unnecessary. We’ve removed the assumption that GPT-4’s parameter count is greater than 175B as part of the above change.
>
> > Performance on this limited dataset may not reflect abilities on more complex real-world tasks
>
> Thank you for raising this point. We agree that high articulation accuracy on ArticulateRules does not correspond to high articulation accuracy on complex real-world tasks. However, we would like to clarify that we’re not claiming that an increase in articulation accuracy on ArticulateRules implies an increase in articulation accuracy on more complex real-world tasks, and this is not the goal of our work. Rather, we’re claiming that high articulation accuracy on this evaluation is a necessary but not sufficient condition for an LLM to be able to articulate its behavior on more complex real-world tasks.
>
> > ”The ArticulateRules dataset is relatively small and simple…”
>
> Thank you for pointing out the need for more exposition regarding our specific dataset construction. We’ve updated the paper to provide more clarity about the design decisions around the size of the dataset and the simplicity of the rules it includes.
>
> ** Size **
>
> The size of the dataset is constrained by compute costs. We assume that a cost of $10’s or $100’s to reproduce our in-context evaluation results is affordable for most labs, and similarly a cost of $1,000’s for our finetuning results. Therefore, if the smallest variant of ArticulateRules exceeded these budgets, it would risk being too costly to run or build upon in practice.
>
> We determined the minimum size of ArticulateRules in the following way: (1) we determined the minimum number of few-shot examples to include in the prompt in order for a human to be able to articulate the rule (we found this to be 16 few-shot examples; more details in Appendix 2.3), (2) we determined the number of examples per rule function needed to have an acceptably-low standard error of the mean for in-context tasks (we found this to be roughly >= 20 and landed on 32 examples per rule function; Figure 11 uses n=15 and has a large SEM, compared to Figure 5 which uses n=32 and has an acceptable SEM to be able to draw conclusions), (3) we then used the maximum number of rule functions given these constraints.
>
> Larger dataset variants are also provided for labs with larger compute budgets. Also, it’s worth noting that the dataset size is easily increased: (1) it is procedurally generated, meaning that larger datasets can be generated as needed, and (2) rule functions are constructed from more primitive ones using AND, OR and NOT operators, meaning it’s straightforward to register more complex rule functions as needed.
>
> ** Simplicity **
>
> We chose tasks where we could: (1) closely approximate the model’s behavior with a known rule (and therefore be able to determine if an articulation was correct or not), and (2) get an accurate picture of the limits of present-day LLMs’ articulation capabilities.
>
> The binary text classification tasks used in ArticulateRules fit both of these criteria. Using easier tasks where LLMs achieved 100% freeform articulation accuracy wouldn’t have given an accurate picture of their limitations. Using harder tasks where LLMs get 0% accuracy would have been equally as uninformative. We tailored these tasks so that present-day LLMs are capable of articulating only a small fraction of the rules in the dataset, which gives a picture of their current articulation capabilities and its limitations, and allows us to track increases in articulation capabilities over time.
>
> We’ve added these details to Appendix B.6 since we agree that the size of the dataset and the complexity of the tasks it includes requires justification.

---

> ### Author Response · Authors · 2023-11-19
> **Responses to your helpful feedback (continued)**
>
> > Only one main model architecture (GPT-3/4) is tested in detail.
>
> Thanks for raising this concern. Our decision to focus on GPT-3/4 was primarily driven by time and cost constraints, however we acknowledge that evaluating LLMs with different architectures would be a valuable experiment to run. In particular, conducting a finetuning evaluation of GPT-3 traded-off against evaluating a wider range of LLMs in-context. We decided that approximating the upper bound of articulation capabilities of GPT-3 (the largest model available with finetuning access) was the more informative experiment to run.
>
> In response to this feedback, we’ve also started evaluating Llama 2 in-context and are planning to have results in the next few days, results of which we will post in a top-level comment.

---

> ### Author Response · Authors · 2023-11-21
>
> Dear Reviewer ZENG, once again thank you for your time and effort in providing your thoughtful review of our paper. As we are now entering the last day or so of the rebuttal period we wanted to just reach out to see if there was any feedback from our response -- we appreciate that this will be a busy time for you but we hope that our current response has addressed any current concerns and are keen to engage further if there are any outstanding concerns. Best wishes, the Authors

---

### Author Response · Authors · 2023-11-19
**Overall response**

We would like to thank all reviewers for their thorough and insightful reviews. We have addressed all reviewer points below, and have made corresponding changes to the paper in **red**. In addition, we are keen to hear if reviewers have any further questions or suggestions to improve the paper.

Reviewers ZENG and ZS9G thought the paper addressed an important topic. Reviewer ZENG also liked the paper because it evaluated models comprehensively and that it showed a major limitation in current LLMs. Reviewer 8rGm thought the paper was well-written and was a valuable contribution.

In response to the weaknesses outlined in the reviews, we’ve made two clarifications to our methodology section: one about the design decisions around the size and complexity of tasks in ArticulateRules to address Reviewer ZENG’s concerns, and another around the assumptions we’re making in Section 2.4 to address Reviewer ZS9G’s concerns. We’ve also removed the assumption about GPT-4’s size being > 175B parameters and amended the claim about articulation accuracy vs model size to address Reviewer ZENG’s concerns.

A common theme was that evaluating an open source model (e.g. Llama) would be a useful contribution. We agree and have started evaluating Llama 2 in-context; we’re planning to have results to share in the next two days, and will post them in reply to this top-level comment when ready.

---

### Author Response · Authors · 2023-11-21

With the discussion period nearing its close, we would like to remind the reviewers that have only left initial reviews that we would appreciate hearing further feedback on our response. We are keen to know if our responses have adequately addressed all their concerns. Should there be any outstanding queries, we are also eager to engage further.

---

### Author Response · Authors · 2023-11-22
**Preliminary results for Llama 2 70B**

We’d like to share preliminary results for Llama 2 70B, which performed poorly on the out-of-distribution classification task and the multiple choice articulation task. However, it performed surprisingly well at the freeform articulation task. We’ve also manually reviewed 30 of the 96 freeform articulations Llama 2 70B generated to give reviewers a sense of our confidence in the freeform articulation results. Below are the following top-3 accuracies for each task (given in the form {mean} +/ {sem}):

- classification (in-distribution): 1.0 +/- 0.0
- classification (out-of-distribution): 0.69 +/- 0.06
- articulation (multiple choice): 0.24 +/- 0.07
- articulation (freeform): 0.42 +/- 0.08

Note that Llama 2 70B’s classification behavior is _not_ well approximated by any of the top three rules (e.g. it’s top score for the out-of-distribution classification task was 0.81 for “contains the word W”), and therefore the freeform articulations are not faithful. (Note: we also find that it either articulates the rule $R$ we expected, or another rule $R'$ inconsistent with the few-shot examples). It also performed worse than all other models on the multiple choice articulation task, since it frequently hallucinated a third-option in the two-way multiple choice task, performing _worse_ than the random baseline.

---

### Meta-Review · Area_Chair_yWQb · 2023-12-11

**Metareview:**

This paper develops a benchmark (ArticulateRules) for evaluating the LLMs capability in explaining their own classification processes. The reviewers agree that this is a timely and important research topic. However, there are several limitations pointed out by the reviewers, and one major one is that the results delivered in the paper with the benchmark is still less convincing. This is caused by the relative small dataset scale and simple rule-based data construction. And it also lacks sufficient in-depth discussion and analysis of why it these models fail on the benchmark.

**Justification For Why Not Higher Score:**

There are several major weaknesses of the paper that prevent it from being accepted into ICLR 2024.

**Justification For Why Not Lower Score:**

N/A

---

### Decision · Program_Chairs · 2024-01-16

Reject